# Characterization of P-Glycoprotein Inhibitors for Evaluating the Effect of P-Glycoprotein on the Intestinal Absorption of Drugs

**DOI:** 10.3390/pharmaceutics13030388

**Published:** 2021-03-15

**Authors:** Yusuke Kono, Iichiro Kawahara, Kohei Shinozaki, Ikuo Nomura, Honoka Marutani, Akira Yamamoto, Takuya Fujita

**Affiliations:** 1Laboratory of Molecular Pharmacokinetics, Graduate School of Pharmaceutical Sciences, Ritsumeikan University, 1-1-1 Noji-Higashi, Kusatsu 525-8577, Japan; y-kono@fc.ritsumei.ac.jp (Y.K.); ph0107ih@ed.ritsumei.ac.jp (H.M.); 2Department of Biopharmaceutics, Kyoto Pharmaceutical University, 5 Misasagi Nakauchi-cho, Yamashina, Kyoto 607-8412, Japan; iichiro.kawahara@jt.com (I.K.); ky02315@poppy.kyoto-phu.ac.jp (K.S.); ky03243@poppy.kyoto-phu.ac.jp (I.N.); yamamoto@mb.kyoto-phu.ac.jp (A.Y.)

**Keywords:** intestinal absorption, P-glycoprotein, breast cancer resistance protein, LY335979, WK-X-34

## Abstract

For developing oral drugs, it is necessary to predict the oral absorption of new chemical entities accurately. However, it is difficult because of the involvement of efflux transporters, including P-glycoprotein (P-gp), in their absorption process. In this study, we conducted a comparative analysis on the inhibitory activities of seven P-gp inhibitors (cyclosporin A, GF120918, LY335979, XR9576, WK-X-34, VX-710, and OC144-093) to evaluate the effect of P-gp on drug absorption. GF120918, LY335979, and XR9576 significantly decreased the basal-to-apical transport of paclitaxel, a P-gp substrate, across Caco-2 cell monolayers. GF120918 also inhibited the basal-to-apical transport of mitoxantrone, a breast cancer resistance protein (BCRP) substrate, in Caco-2 cells, whereas LY335979 hardly affected the mitoxantrone transport. In addition, the absorption rate of paclitaxel after oral administration in wild-type mice was significantly increased by pretreatment with LY335979, and it was similar to that in *mdr1a/1b* knockout mice. Moreover, the absorption rate of topotecan, a BCRP substrate, in wild-type mice pretreated with LY335979 was similar to that in *mdr1a/1b* knockout mice but significantly lower than that in *bcrp* knockout mice. These results indicate that LY335979 has a selective inhibitory activity for P-gp, and would be useful for evaluating the contribution of P-gp to drug absorption.

## 1. Introduction

Most new chemical entities (NCEs) are developed as oral drug formulations because the oral route has several advantages, such as its convenience, non-invasiveness, and good patient adherence [1]. However, a lot of NCEs suffer from poor intestinal permeability because they are recognized and transported by efflux transporters, such as P-glycoprotein (P-gp) and breast cancer resistant protein (BCRP), expressed on the apical membrane of small intestinal epithelial cells [2].

P-glycoprotein (P-gp) is a member of the ATP-binding cassette (ABC) transporter family and exhibits broad substrate specificity. P-gp substrates tend to have a large molecular volume, electronegative groups, and hydrogen bonding groups [3], and a large number of drugs, including anti-cancer drugs and steroids, have been identified as substrates for P-gp [4]. The intestinal permeability of P-gp substrates is known to be lower than that estimated from their lipophilicity [5]. On the other hand, several drugs, such as verapamil and quinidine, are efficiently absorbed from the small intestine, although they are typical substrates for P-gp [6]. Therefore, it is important to appropriately assess the contribution of P-gp to the intestinal permeability of NCEs in order to evaluate their oral absorption.

For evaluating the effect of P-gp on the intestinal absorption of drugs, Caco-2 cells, a human colorectal adenocarcinoma cell line, are widely used in vitro studies because Caco-2 cells express several solute carrier (SLC) transporters, ABC transporters, and metabolic enzymes [7,8,9]. On the other hand, *mdr1a/1b* knockout (KO) mice are often used for in vivo investigations of P-gp function [10,11,12]. However, there is little evidence that the physiological function of the gastrointestinal tract in *mdr1a/1b* KO mice is as same as that in wild-type (WT) mice. Therefore, it is necessary to develop a new approach to assess the in vivo contribution of P-gp to drug absorption correctly and conveniently, instead of using *mdr1a/1b* KO mice.

Previous reports have demonstrated that the effect of P-gp on in vivo drug absorption could evaluate in WT animals by using P-gp inhibitors, such as cyclosporin A (CsA) and verapamil [13,14]. However, other efflux transporters such as BCRP are expressed in the intestinal epithelial cells [15]. Moreover, the substrate specificity of P-gp extensively overlaps with that of cytochrome P450 (CYP) 3A [16], and the metabolism and elimination of these substrates have been reported to be conducted concertedly [17]. Taking the above into consideration, P-gp inhibitors would inhibit the function of not only P-gp but other efflux transporters and metabolic enzymes, and therefore P-gp inhibitors require the excellent affinity and selectivity for P-gp.

So far, various P-gp inhibitors have been developed for overcoming multidrug resistance of cancer [18,19,20]. First-generation P-gp inhibitors, CsA and verapamil, have pharmacological activities with low affinity and low transporter selectivity [18,19]. Since verapamil is a well-known substrate for P-gp, verapamil inhibits the function of P-gp in a competitive manner [21]. Second-generation P-gp inhibitors, including VX-710 (biricodar), lack the same pharmacological activity, and they have relatively high and selective inhibitory activity on P-gp. However, these inhibitors also inhibit CYP3A4 [18,19]. Third-generation P-gp inhibitors, including GF120918 (elacridar), LY335979 (zosuquidar), XR9576 (tariquidar), WK-X-34, and OC144-093 (ontogen), are capable of inhibiting P-gp function at a lower concentration compared with first- and second-generation P-gp inhibitors. In addition, their affinity for CYP3A4 is lower than second-generation inhibitors [18,19,20]. These inhibitors have been reported to improve the oral bioavailability (BA) and area under the curve (*AUC*) for plasma concentration-time of P-gp substrates [22,23,24]. However, there are few reports performing the comparative analysis of these P-gp inhibitors to select suitable ones for evaluating the effect of P-gp on intestinal drug absorption.

In this study, we selected seven P-gp inhibitors, CsA, GF120918, XR9576, LY335979, WK-X-34, VX-710, and OC144-093 (Figure 1), and comparatively evaluated their inhibitory activity and selectivity for P-gp both in vitro and in vivo. Here, we used paclitaxel as a well-known P-gp substrate with no affinity for BCRP for both in vitro and in vivo studies [25]. We also used mitoxantrone as a typical substrate for BCRP with a lower affinity for P-gp for in vitro studies [26]. For in vivo studies, topotecan, which we have used in pharmacokinetic studies in mice, was selected as a typical BCRP substrate [12,27].

## 2. Materials and Methods

### 2.1. Chemicals

*N*-(4-(2-(6,7-Dimethoxy-3,4-dihydroisoquinolin-2(1*H*)-yl)ethyl)phenyl)-5-methoxy-9-oxo-9,10-dihydroacridine-4-carboxamide (GF120918), *N*-(2-([(4-(2-[3,4-Dihydro-6,7-dimethoxy-2(1H)-isoquinolinyl]ethyl)phenyl)amino]carbonyl)-4,5-dimethoxyphenyl)-3-quinolinecarboxamide (XR9576), paclitaxel, mitoxantrone, Hank’s Balanced Salt (HBS) without phenol red and sodium bicarbonate, and solutol were obtained from Sigma-Aldrich (St. Louis, MO, USA). (*R*)-1-(4-([1a*R*,6s,10b*S*]-1,1-difluoro-1,1a,6,10b-tetrahydrodibenzo[a,e]cyclopropa(c)(7)annulen-6-yl)piperazin-1-yl)-3-(quinolin-5-yloxy)propan-2-ol trihydrochloride (LY335979) was purchased from Funakoshi (Tokyo, Japan). *N*-(2-(4-[2-(6,7-Dimethoxy-3,4-dihydro-1*H*-isochinolin-2-yl)-ethyl]phenylcarbamoyl)phenyl)-3,4-dimethoxybenzamide (WK-X-34) was purchased from Cosmo Bio Co. Ltd. (Tokyo, Japan). (*S*)*-N-*(2-Oxo-2-(3,4,5-trimethoxyphenyl)-acetyl)piperidine-2-carboxylic acid 1,7-bis(3-pyridyl)-4-heptyl ester (VX-710) was obtained from Namiki Shoji Co. Ltd. (Tokyo, Japan). (*E*)-4,4’-(2-(4-[3-ethoxyprop-1-en-1-yl]phenyl)-1H-imidazole-4,5-diyl)bis(*N*-isopropylaniline) (OC144-093) was purchased from MedKoo Biosciences, Inc. (Morrisville, NC, USA). Dulbecco’s Modified Eagles Medium (DMEM), antibiotic-antimycotic mixed stock solution (×100), 0.25% trypsin/1 mM EDTA solution, nonessential amino acids, and l-glutamine were purchased from Nacalai Tesque (Kyoto, Japan). Fetal bovine serum was purchased from Thermo Fisher Scientific (Waltham, MA, USA). CsA was purchased from Tokyo Chemical Industry Co., Ltd. (Tokyo, Japan). [^3^H]Taxol (identical to paclitaxel, specific radioactivity: 12.9 Ci/mmol) was purchased from Moravek Biochemicals (Brea, CA, USA). Transwell^®^ was purchased from Corning (Corning, NY, USA). Topotecan HCl was purchased from ALEXIS CORPORATION (Lausen, Switzerland). WellSolve, a water-soluble solubilizing agent [30], was kindly supplied from Celeste Corporation (Tokyo, Japan). Other chemicals were all of guaranteed reagent grade and were obtained commercially.

### 2.2. Cell Culture

Caco-2 cells from Dainippon Sumitomo Pharma (Osaka, Japan) were cultured in DMEM supplemented with 10% heat-inactivated fetal bovine serum, 1% antibiotic-antimycotic, 1% nonessential amino acids, and 2 mM l-glutamine [31]. Cells were maintained at 37 °C in 5% CO_2_/95% air and passaged upon reaching approximately 80% confluence using 0.25% trypsin/1 mM EDTA solution. For transport experiments, Caco-2 cells were grown on a polyethylene terephthalate insert (0.4 μm pore size, 12 mm diameter) at 1.0 × 10^5^ cells/well, and cultured for 14–21 days with the replacement of the medium once every 2 days. Caco-2 cells used in this study were within the passage range of 55 through 70.

### 2.3. Animals

Male WT FVB mice (20–30 g, 8 weeks old), and *mdr1a/1b*^−/−^ (mdr1a/1b KO) and *bcrp*^−/−^ (bcrp KO) mice of the same genetic background (FVB) mice (20–30 g, 8 weeks old) were obtained from Taconic Farms (Germantown, NY, USA). Animals were maintained under standard conditions (temperature at 23 ± 1 °C with a relative humidity between 40% and 60%) with a 12 h light/dark cycle. Food and water were available *ad libitum*. All animal experiments were carried out in accordance with principles and procedures outlined in the National Institutes of Health Guide for the Care and Use of Laboratory. All experimental animal procedures were reviewed and approved by the Animal Care and Use Committee of the Kyoto Pharmaceutical University (2005-239) and Ritsumeikan University (BKC2010-27).

### 2.4. Transport/Inhibition Experiments

Each P-gp inhibitor was dissolved in dimethylsulfoxide (DMSO) at concentrations of 0.002, 0.006, 0.02, 0.06, 0.2, 0.6, and 2 mM, and diluted 200 times with HBS solution (HBSS) at pH 6.5 or pH 7.4. The final % of DMSO in HBSS was 0.5%. The confluent Caco-2 cells were washed with HBSS, and the transepithelial electrical resistance (TEER) values of the cell monolayer were measured using a Millicell-ERS volt-ohm meter (EMD Millipore Co., MA, USA). Then, 0.5 mL of HBSS at pH 6.5 with 0.5% DMSO in the presence or absence of each P-gp inhibitor (0.01, 0.03, 0.1, 0.3, 1, 3, and 10 μM) was added to the apical (AP) side of the cell monolayer. Similarly, 1.5 mL of HBSS at pH 7.4 with 0.5% DMSO in the presence or absence of each P-gp inhibitor was added to the basal (BL) side of the cell monolayer. After preincubating for 10 min, either AP or BL side of the cell monolayer was replaced with HBSS at pH 6.5 or pH 7.4, containing each inhibitor and paclitaxel (5 μM) or mitoxantrone (5 μM) [32,33]. Small amounts of paclitaxel and mitoxantrone were replaced with [^3^H]taxol and [^3^H]mitoxantrone, respectively. Lucifer yellow (100 μM) was also added to either the AP or BL side as a paracellular marker. The cells were incubated at 37 °C, and 100 μL of the medium was collected from each compartment at specified times. After the sample collection, an equal volume of fresh HBSS at pH 6.5 or pH 7.4 with 0.5% DMSO containing each inhibitor was immediately added to each compartment. After the transport experiments, the medium was collected from the donor side for measuring the drug recovery, and the TEER values of the cell monolayer were measured again.

### 2.5. Preparation of Drug Solution for In Vivo Study

For oral administration, paclitaxel was dissolved at a concentration of 2 mg/mL in water with 1% DMSO and 20% WellSolve. Topotecan was dissolved at a concentration of 2.5 mg/mL in water with 1% DMSO and 10% Solutol. For intravenous injection, paclitaxel was dissolved at a concentration of 1 mg/mL in water with 1% DMSO and 20% WellSolve. The solution was filtrated through a 0.22 μm sterile membrane filter (EMD Millipore Co., Tokyo, Japan).

The in vivo pharmacokinetic studies were carried out according to the portal-systemic blood concentration (P-S) difference method [12,34,35]. The mice (*n* = 3) were orally administered with WK-X-34 or LY335979 at a dose of 40 mg/kg or 60 mg/kg, respectively. At 10 min after the administration of P-gp inhibitors, paclitaxel was orally administered at a dose of 20 mg/kg. In another experiment, the mice (*n* = 3) were intravenously administered with PTX via the tail vein at a dose of 5 mg/kg. These doses of PTX are less than those used in the previous reports [36,37]. After the administration, 300 μL of blood samples were collected from the portal and abdominal veins, respectively, of the mice under isoflurane anesthesia at 0.083, 0.17, 0.5, 1, 2, 4, and 8 h. Three mice per group were used at each time point, and the mice were euthanized after the sample collection. For evaluating the absorption rate, the mice (*n* = 2–6) were orally administered with CsA, WK-X-34, or LY335979 at a dose of 15–30 mg/kg. Then, paclitaxel or topotecan was orally administered at a dose of 10 mg/kg or 2 mg/kg, respectively [36,38]. After the administration, 300 μL of blood samples were collected from the portal and abdominal veins, respectively, of the mice under isoflurane anesthesia at 10 or 30 min. Two to six mice per group were used at each time point, and the mice were euthanized after the sample collection. The collected blood samples were immediately centrifuged at 14,000 g for 10 min at 4 °C, and the plasma samples were obtained. The plasma samples were kept at −80 °C until sample analysis.

### 2.6. Analytical Methods

The radiolabeled compounds ([^3^H]taxol and [^3^H]mitoxantrone) were measured by mixing the samples with a scintillation cocktail (Clearsol I; Nacalai Tesque) in counting vials, followed by placing them in a liquid scintillation counter (LS-6500; Beckman Instruments, Fullerton, CA, USA). The concentration of lucifer yellow was determined by measuring fluorescent intensity at a wavelength of 428 nm (Excitation (Ex))/540 nm (Emission (EM)) using an Infinite F200 microplate reader (Tecan Japan Co. Ltd. Kanagawa, Japan).

In animal studies, 0.1 mL of plasma samples were mixed with 0.9 mL of ultra-pure water and 6 mL of ethyl acetate for extracting paclitaxel. For extracting topotecan, 0.1 mL of the samples were mixed with 0.1 mL of 0.85% phosphoric acid and 1 mL of acetonitrile. The mixture was centrifuged at 750 g for 10 min at 4 °C, and the supernatants were collected. After the evaporation of the supernatants, the residues were dissolved in the high-performance liquid chromatography (HPLC) mobile phase. Paclitaxel and topotecan were measured using HPLC (Shimadzu LC-10AS pump, SIL-10A autosampler; Shimadzu) equipped with a reverse-phase column (COSMOSIL 5C_18_-AR-II, 3.5 μm inner diameter, 4.6 × 100 mm). The composition of the mobile phase for paclitaxel was 20 mM potassium phosphate buffer (pH 3.0) with acetonitrile, according to the following gradient program:0–15.0 min, 45% acetonitrile15.0–25.0 min, 45–70% acetonitrile25.0–30.0 min, 70% acetonitrile30.0–40.0 min, 70–45% acetonitrile40.0–50.0 min, 45% acetonitrile

The composition of the mobile phase for topotecan was 10 mM phosphate buffer (pH 3.74) with methanol (76:24, *v*/*v*). The flow rate was 1.0 mL/min. Paclitaxel was detected by absorbance at 227 nm using Shimadzu SPD-20A UV spectrophotometric detector. Topotecan was analyzed by measuring fluorescence (Excitation: 361 nm, Emission: 527 nm) with a Shimadzu RF-10A XL fluorescence detector.

### 2.7. Data Analysis

Pharmacokinetic data analysis was performed using SigmaPlot software (HULINKS Inc., Tokyo, Japan).

For in vitro transport studies, mass balance (% recovery) was calculated using Equation (1)
% recovery = (*C*_D,4 h_ × *V*_D_ + *C*_R,4 h_ × *V*_R_)/*C*_D,0 h_ × *V*_D_(1)
where *C*_D,0 h_ is the initial drug concentration at the donor side, *C*_D,4 h_ and *C*_R,4 h_ are the drug concentration at the donor and receiver side, respectively, at 4 h; *V*_D_ and *V*_R_ are the solution volumes at the donor and receiver side, respectively.

The apparent permeability coefficient (*P*_app_) was calculated using Equation (2)
*P*_app_ = Δ*Q*/Δ*t* × 1/(A × *C*_0_)(2)
where Δ*Q*/Δ*t* is the transported flux of paclitaxel or mitoxantrone, A is the surface area of the porous membrane (1.13 cm^2^), and *C*_0_ is the initial concentration of paclitaxel or mitoxantrone added to the donor compartment.

The efflux ratio (*ER*) was calculated using Equation (3)
*ER* = *P*_app,BA_/*P*_app,AB_(3)
where *P*_app,AB_ and *P*_app,BA_ are the *P*_app_ values for AP-to-BL and BL-to-AP transport, respectively.

The inhibitor concentration to achieve 50% increase of *P*_app,AB_ of paclitaxel or mitoxantrone (*IC*_50_) was obtained by fitting the collected permeability data to the Equation (4)
*P*_app,AB_ = Range/[1 + (*C*/*IC*_50_)^γ^] + Background (4)
where *C* is the concentration of an inhibitor, γ is the Hill coefficient, Range is the arithmetic difference of *P*_app,AB_ value between on complete inhibition and in the absence of an inhibitor; Background is the *P*_app,AB_ in the absence of inhibitors. *IC*_50_ values of *P*_app,BA_ and *ER* were also determined using Equation (4).

For in vivo pharmacokinetic studies, apparent *F*_a_*F*_g_ (*F*_a_, absorption ratio; *F*_g_, intestinal availability) in the P-S difference method was calculated by Equation (5)
*F*_a_*F*_g_ = *Q*_pv_ × *R*_b_ × (*AUC*_pv_ − *AUC*_sys_)/*Dose*(5)
where *Q*_pv_ is the portal blood flow (72.5 mL/min/kg) [39], *R*_b_ is the blood/plasma concentration ratio, *AUC*_pv_ and *AUC*_sys_ are the *AUC* in the portal vein, and systemic circulation after oral administration, respectively.

The apparent absorption rate (*V*) was estimated using Equation (6)
*V* = *Q*_pv_ × *R*_b_ × (*C*_pv_ − *C*_sys_)(6)
where *C*_pv_ and *C*_sys_ are the drug concentrations in the portal vein and systemic circulation, respectively.

The elimination rate constant (*k*_e_) was determined by using least-squares regression analysis of plasma concentration versus time curve. Elimination half-life (*t*_1/2_) was calculated using Equation (7):*t*_1/2_ = ln2/*k*_e_(7)

*AUC* and area under the first moment curve (*AUMC*) from time 0 to infinity were calculated using the trapezoidal rule. Mean residence time (*MRT*), total body clearance (*CL*_tot_), and distribution volume at the steady state (*Vd*_ss_) were calculated using the following equations:*MRT* = *AUMC*/*AUC*(8)
*CL*_tot_ = *Dose*/*AUC*(9)
*Vd*_ss_ = *MRT* × *CL*_tot_(10)

Mean absorption time (*MAT*) and absorption constant (*k*_a_) after oral administration were calculated using the following equations
*MAT* = *MRT*_oral_ − *MRT*_iv_(11)
*k*_a_ = 1/*MAT*(12)
where *MRT*_oral_ and *MRT*_iv_ are the *MRT* after oral and intravenous administration, respectively.

*BA* was calculated by Equation (13)
*BA* = *AUC*_sys_/*AUC*_iv_ × *Dose*_iv_/*Dose*_oral_ × 100(13)
where *AUC*_iv_ is the *AUC* after intravenous administration. *Dose*_iv_ and *Dose*_oral_ are administered doses in the intravenous and oral administration study, respectively.

Hepatic availability (*F*_h_) was calculated by Equation (14):*F*_h_ = *BA*/(*F*_a_ × *F*_g_)(14)

### 2.8. Statistical Analysis

In vitro transport experiments were carried out in 3 independent cell passages, ranging from 55 to 70. Here, data about in vitro transport experiments are represented as the mean ± standard deviation (S.D.) for 3 experiments using Caco-2 cells with different passages with 3 replicates per *n*. For the in vivo study, 2–6 mice were used in each group, and data obtained from more than 3 mice were represented as the mean ± S.D.

## 3. Results

### 3.1. Inhibitory Effect of P-gp Inhibitors on the Transport of Paclitaxel in Caco-2 Cells

Prior to the transport study, we measured the mRNA expression levels of efflux transporters in Caco-2 cells. We could detect the mRNA expression of MDR1 and BCRP in Caco-2 cells, although their expression levels were lower than those in human small intestine cells (Appendix A). We also confirmed the barrier properties of Caco-2 cell monolayers. The transepithelial electrical resistance value of the monolayer was more than 500 Ω·cm^2^, and it was maintained during the transport study. In addition, the *P*_app,AB_ value of lucifer yellow, a robust paracellular permeability marker, was very low (0.10 × 10^−6^ cm/s). These results indicate that the tight junction in Caco-2 cell monolayers was effectively formed and maintained during the experiments. In addition, the recovery of paclitaxel in all transport studies was 85–105%. Then, we evaluated the inhibitory effects of P-gp inhibitors on the P-gp-mediated efflux of paclitaxel across Caco-2 cells. The seven P-gp inhibitors used in this study are listed in Figure 1. We observed that the AP-to-BL transport of paclitaxel was increased by the presence of P-gp inhibitors, except for OC144-093, in a concentration-dependent manner (Figure 2, Appendix A). On the other hand, the BL-to-AP transport of paclitaxel was significantly decreased by increasing inhibitor concentrations. Table 1 summarizes the *IC*_50_ values and Hill coefficients of each inhibitor for both AP-to-BL and BL-to-AP transport of paclitaxel across Caco-2 cell monolayers. GF120918, XR9576, LY335979, and WK-X-34 had much lower *IC*_50_ values than CsA. These results indicate that GF120918, XR9576, and LY335979 have potent inhibitory activities on P-gp. On the other hand, the *IC*_50_ value of VX-710 was higher than that of CsA. In the case of OC144-093, the transport of paclitaxel in Caco-2 cells was not changed by its addition, indicating that OC144-093 is not a suitable P-gp inhibitor.

### 3.2. Effect of P-gp Inhibitors on BCRP-Mediated Drug Efflux in Caco-2 Cells

In order to determine the selectivity of P-gp inhibitors, we investigated the effect of P-gp inhibitors on the transport of mitoxantrone, a typical BCRP substrate, across Caco-2 cells monolayers. The recovery of mitoxantrone in all transport studies was 80–85%. The AP-to-BL transport of mitoxantrone was hardly affected by increasing the concentration of P-gp inhibitors (Figure 3 and Appendix A). On the contrary, six P-gp inhibitors (not LY335979) significantly decreased the BL-to-AP transport of mitoxantrone across Caco-2 cell monolayers (Figure 3 and Appendix A). The *IC*_50_ value and Hill coefficients of these compounds are summarized in Table 2. The *IC*_50_ values of GF120918, WK-X-34, and VX-710 were similar to those cases in paclitaxel (239 nM, 501 nM, and 4496 nM, respectively), indicating that these compounds seem to act as dual inhibitors for P-gp and BCRP. On the other hand, the inhibitory activity of XR9576 and LY335979 on the BL-to-AP transport of mitoxantrone values were lower than those of paclitaxel (Table 2). These results indicate that XR9576 and LY335979, particularly LY335979, seem to be selective inhibitors for P-gp.

### 3.3. In Vivo Inhibitory Effect of LY335979 and WK-X-34 on P-gp- and BCRP-Mediated Drug Efflux

Since LY335979 is shown to be a potent and selective inhibitor for P-gp, we next evaluated the in vivo effect of LY335979 on the intestinal absorption of paclitaxel in mice by using the P-S difference method. In addition to LY335979, we also investigated the effect of WK-X-34 as a dual inhibitor for P-gp and BCRP.

Initially, we investigated the contribution of P-gp to the intestinal absorption of paclitaxel in mice. As shown in Figure 4A,B and Table 3, the *AUC*_sys_ value of paclitaxel after oral administration in *mdr1a/1b* KO mice was 3086 nM∙h, which was approximately 2.8-fold higher than that in WT mice (1089 nM∙h). Moreover, the *AUC*_pv_ value in *mdr1a/1b* KO mice was 3.3-fold as high as that in WT mice. The maximum plasma concentration (*C*_max_) value of paclitaxel in systemic circulation and portal vein after oral administration in *mdr1a/1b* KO mice was 1179 nM and 2523 nM, respectively, and each value was 2.7- and 3.5-fold higher than that in WT mice (442 nM and 730 nM, respectively). On the other hand, the *AUC*_iv_, *Vd*_ss_, and the *CL*_tot_ of paclitaxel after intravenous injection were not significantly different between WT mice and *mdr1a/1b* KO mice. Based on these results, we calculated the *BA* and *F*_a_*F*_g_ values of paclitaxel in *mdr1a/1b* KO mice (29.0% and 66.3%, respectively), and these values were 2.5- and 4.0-fold as high as those in WT mice (11.7% and 16.6%, respectively). These results indicate that the intestinal absorption of paclitaxel is restricted by P-gp expressed in the intestinal epithelium.

We next evaluated the effect of P-gp inhibitors on the oral absorption of paclitaxel in mice. Figure 4C illustrates the plasma concentration profile of paclitaxel (20 mg/kg) after oral administration in WT mice pretreated with WK-X-34 (40 mg/kg). The calculated pharmacokinetic parameters are listed in Table 3. The *k*_e_ value of paclitaxel in systemic circulation after oral administration in WT mice pretreated with WK-X-34 (0.23 h^−1^) was much lower than that in non-treated WT mice (0.50 h^−1^). Moreover, it was also lower than that in non-treated *mdr1a/1b* KO mice (0.35 h^−1^). In addition, the *AUC*_sys_ value of paclitaxel in WT mice pretreated with WK-X-34 was 8636 nM∙h, which was approximately three times higher than that in non-treated *mdr1a/1b* KO mice (3086 nM∙h). These results suggest that WK-X-34 inhibits transporters, apart from P-gp, or metabolic enzymes involved in the elimination process of paclitaxel.

We also assessed the pharmacokinetic profile of paclitaxel (20 mg/kg) after oral administration in WT or *mdr1a/1b* KO mice with or without LY335979 pretreatment (60 mg/kg). We preliminarily confirmed that this dose of LY335979 is enough to potently inhibit P-gp in the murine small intestine, and this dose is reasonable because the previous report used LY335979 at a dose of 25–80 mg/kg [40]. The *C*_max_ value of paclitaxel in systemic circulation after oral administration in WT mice pretreated with LY335979 was 1208 nM, which was approximately 2.7-fold higher than that in non-treated WT mice (442 nM) and as same as that in non-treated *mdr1a/1b* KO mice (1179 nM) (Figure 4D, Table 3). The *C*_max_ value of paclitaxel in portal vein in LY335979-pretreated WT mice was also similar to that in non-treated *mdr1a/1b* KO mice. However, the *AUC*_sys_ and *AUC*_pv_ values of paclitaxel after oral administration in WT mice pretreated with LY335979 were 3201 nM∙h and 7970 nM∙h, respectively, and these were slightly higher than those in *mdr1a/1b* KO mice (3086 nM∙h and 6575 nM∙h, respectively). Furthermore, the *F*_a_*F*_g_ value in WT mice pretreated with LY335979 was 86.8%, which was approximately 5.2- and 1.3-fold higher than that in non-treated WT mice and *mdr1a/1b* KO mice, respectively (16.6% and 66.3%, respectively). These results suggest that LY335979 has the potential to inhibit the metabolic enzymes in intestinal epithelial cells. On the other hand, the *k*_e_ value of paclitaxel in systemic circulation after oral administration in WT mice pretreated with LY335979 (0.64 h^−1^) was not lower than that in non-treated WT mice and *mdr1a/1b* KO mice (0.50 and 0.35 h^−1^). These results suggest that LY335979 hardly affects the metabolic enzymes involved in the elimination process of paclitaxel.

### 3.4. Effect of P-gp Inhibitors on the Absorption Rate of Paclitaxel

We have evaluated the effect of P-gp inhibitors on the intestinal absorption of paclitaxel in mice by using P-S difference method; however, there was a risk of overestimation or underestimation of the pharmacokinetics of paclitaxel because of the duration of P-gp inhibitors and the effect of inhibitors on metabolic enzymes involved in absorptive and/or elimination process of paclitaxel. Therefore, in order to minimize the effect of P-gp inhibitors on factors except for P-gp, we next investigated the absorption rate (*V*) of paclitaxel in mice by P-S difference method with two improvements; to decrease a dose of P-gp inhibitors and to carry out the blood sampling at early time points. In this experiment, a dose of paclitaxel was also decreased to 10 mg/kg, along with the reduction in a dose of P-gp inhibitors.

Table 4 shows the plasma concentration of paclitaxel (10 mg/kg) in systemic circulation and portal vein at 10 min after oral absorption in mice pretreated with P-gp inhibitors, and the absorption rates were also calculated. The *C*_sys_ value was not significantly different between WT mice and *mdr1a/1b* KO mice. However, the *C*_pv_ value in *mdr1a/1b* KO mice was 967 nM, which was higher than that in WT mice (439 nM). Moreover, the *V* value in mdr1a/1b mice (54.3 nmol/min/kg) was approximately 3.5-fold higher than that in WT mice (15.7 nmol/min/kg). When WT mice were pretreated with CsA and WK-X-34, the *V* value of paclitaxel was increased to 38.7 nmol/min/kg and 39.8 nmol/min/kg, respectively, which were slightly lower than that in *mdr1a/1b* KO mice. On the other hand, the *V* value in WT mice was significantly increased by the pretreatment with LY335979 (60.2 nmol/min/kg). This *V* value was as same as that in WT mice. In addition, the *C*_sys_ value in WT mice pretreated with CsA, LY335979, and WK-X-34 was similar to that in *mdr1a/1b* KO mice. Thus, we succeeded in evaluating the effect of P-gp inhibitors on the absorption of paclitaxel with minimizing their influence on the metabolism.

We also assessed the plasma concentration of paclitaxel (10 mg/kg) in systemic circulation and in the portal vein and its absorption rate at 30 min after oral absorption in mice pretreated with P-gp inhibitors (Table 5). The *C*_sys_ value in *mdr1a/1b* KO mice was slightly higher than that in WT mice, whereas the *C*_pv_ value in *mdr1a/1b* KO mice was significantly higher than that in WT mice. Consequently, the *V* value in *mdr1a/1b* KO mice was 63.7 nmol/min/kg, which was approximately 5.1-fold higher than that in WT mice (12.4 nmol/min/kg). Although the *C*_sys_ and *C*_pv_ values in WT mice were increased by the pretreatment with P-gp inhibitors at a concentration of 15 mg/kg, the differences were not remarkable. On the other hand, the *C*_sys_, *C*_pv_, and *V* values in WT mice were all increased significantly when the mice were pretreated with 30 mg/kg of P-gp inhibitors. In particular, the *V* values in WT mice were increased to 66.4 nmol/min/kg by the pretreatment with LY335979, which was almost as same as that in *mdr1a/1b* KO mice. Meanwhile, the *V* values in CsA-pretreated WT mice were lower than that in *mdr1a/1b* KO mice. Although the *C*_sys_ value of paclitaxel in WT mice pretreated with 30 mg/kg of P-gp inhibitors was higher than that in non-treated WT mice and *mdr1a/1b* KO mice, LY335979 exhibited the least effect on the *C*_sys_. These results suggest that LY335979 has the least effect on the metabolic enzymes for paclitaxel even at 30 min after oral administration of paclitaxel. On the other hand, the *V* and *C*_sys_ values in WK-X-34-pretreated WT mice were higher than that in *mdr1a/1b* KO mice, suggesting that WK-X-34 greatly affects the metabolic process of paclitaxel.

### 3.5. Effect of P-gp Inhibitors on BCRP-Mediated Efflux In Vivo

In order to determine the selectivity of P-gp inhibitors for p-gp in vivo, we also assessed the effect of P-gp inhibitors on the *C*_sys_, *C*_pv_, and *V* values of topotecan, a substrate for BCRP, at 30 min after oral administration in mice. As shown in Table 6, the *C*_sys_ value of topotecan in *mdr1a/1b* KO mice was similar to that in WT mice, whereas the *C*_sys_ value in *bcrp* KO mice was higher than that in WT mice and *mdr1a/1b* KO mice. Moreover, the *C*_pv_ value in *bcrp* KO mice was 469 nM, and it was approximately twice as high as that in WT mice and *mdr1a/1b* KO mice. Furthermore, the *V* value of topotecan in *bcrp* KO mice was 5.30 nmol/min/kg, which was approximately 4.1-fold and 2.4-fold higher than that in WT mice and *mdr1a/1b* KO mice, respectively. These results indicate that BCRP greatly influences the intestinal absorption of topotecan. We also observed that the *V* value of topotecan in WT mice was significantly increased by the pretreatment with 30 mg/kg of WK-X-34, and reached the same level as in *bcrp* KO mice. On the other hand, LY335979 showed little effect on the *V* value of topotecan in WT mice, and this value was as same as that in *mdr1a/1b* KO mice. These results suggest that WK-X 34 has a potent inhibitory activity for BCRP, whereas LY335979 has little affinity for BCRP.

## 4. Discussion

In this study, we assessed the usability of P-gp inhibitors for evaluating the influence of P-gp on intestinal absorption of drugs both in vitro and in vivo.

In vitro transport studies, we determined the *P*_app,AB_, and *P*_app,BA_ of paclitaxel and mitoxantrone in Caco-2 cells with or without P-gp inhibitors. We observed that the *IC*_50_ value of each P-gp inhibitor for *P*_app,AB_ of paclitaxel was higher than that for *P*_app,BA_ of paclitaxel. In addition, the increase in the *P*_app,AB_ of mitoxantrone by the presence of P-gp inhibitors was hardly observed, and consequently, the *IC*_50_ of P-gp inhibitors for *P*_app,AB_ of mitoxantrone could not be determined. These results suggest that the inhibitory efficiency of P-gp inhibitors against P-gp- and BCRP-mediated drug transport is lower in absorptive direction than in secretory direction. Troutman et al. have demonstrated that apparent *K*_m_ for P-gp-mediated efflux of P-gp substrates in Caco-2 cells was much larger in absorptive direction than in secretory direction [41]. This finding is in accordance with our present observation. In addition, we also considered that the difference in the inhibitory activity of P-gp inhibitors between absorptive direction and secretory direction would be responsible for the inaccurate *ER* values (Appendix A). Therefore, we decided to evaluate the inhibitory effects of P-gp inhibitors on P-gp and BCRP based on the *IC*_50_ for *P*_app,BA_ of paclitaxel, and mitoxantrone.

In Caco-2 transport studies, we observed that GF120918, WK-X-34, and VX-710 had the inhibitory effect for both P-gp and BCRP. Jonker et al. demonstrated that the *AUC* of topotecan after oral administration in *mdr1a/1b* KO mice treated with GF120918 was approximately six-fold higher than that in non-treated mice [42]. Other studies have also reported that GF120918 significantly increases the plasma and brain concentration of dasatinib and crizotinib, which is a substrate for P-gp and BCRP, respectively, in WT mice up to the equal level in *mdr1a/1b* or *bcrp* KO mice [43,44]. These findings indicate that GF120918 is a potent dual inhibitor for P-gp and BCRP. WK-X-34 has also been reported to significantly inhibit the cellular uptake of mitoxantrone in BCRP-overexpressing MCF7 cells [20]. Since GF120918 and WK-X-34 have similar chemical structures, including N-ethyl-tetrahydroisoquinoline, which frequently appears in potent P-gp inhibitors [45], this structure would have the potential to be recognized by BCRP. VX-710 has also been shown to be capable of inhibiting the uptake of mitoxantrone and SN-38 into BCRP-overexpressing MCF7 AdVp3000 cells [46]. These findings support the present results. Taken together, these three P-gp inhibitors, especially GF120918 and WK-X-34, would be useful for evaluating the pharmacokinetics of drugs under conditions of inhibiting the efflux transporters.

We also revealed that XR9576 and LY335979 strongly inhibited P-gp whereas their affinity for BCRP was relatively low in Caco-2 permeation studies. In particular, the *IC*_50_ value of LY335979 for *P*_app,BA_ of paclitaxel was less than one hundredth lower than that of mitoxantrone. These results agree with the report of Shepard et al., demonstrating that the affinity of LY335979 for P-gp is 100-fold higher than that for BCRP [47]. On the other hand, the *IC*_50_ value of XR9576 for *P*_app,BA_ of paclitaxel was only 15.6-fold lower than that of mitoxantrone. Pick et al. have reported that the IC_50_ value of XR9576 for P-gp-mediated Hoechst 33342 transport is approximately 20.1-fold lower than that for BCRP-mediated Hoechst 33342 transport [48]. These observations are in accordance with our present results. To summarize these findings, XR9576 and LY335979, particularly LY335979, were found to be potent and selective P-gp inhibitors.

Then, we evaluated the effect of WK-X-34 and LY335979 as a dual inhibitor and P-gp-selective inhibitor, respectively, on the intestinal absorption of paclitaxel in vivo. So far, several P-gp inhibitors have been reported to be evaluated their effects on the intestinal absorption of paclitaxel in vivo [35,49,50]. However, most of those studies determined the pharmacokinetic parameters of paclitaxel from the plasma concentration in systemic circulation. On the other hand, here, we applied the P-S difference method to determine the pharmacokinetic parameters of paclitaxel because this method would be useful for obtaining the exact *F*_a_*F*_g_ values.

We observed that WK-X-34 delayed the elimination of paclitaxel, suggesting that WK-X-34 inhibits not only efflux transporters, including P-gp and BCRP, but also metabolic enzymes. It has been reported that paclitaxel is mainly metabolized by CYP2C8 and CYP3A4 [51,52]. In addition, we also confirmed that the *AUC*_pv_ of paclitaxel after oral administration in WT mice pretreated with WK-X-34 was only 1.6-fold higher than that in non-treated *mdr1a/1b* KO mice, and the *F*_a_*F*_g_ value in WT mice pretreated with WK-X-34 was only half as high as that in non-treated *mdr1a/1b* KO mice. This may be due to the incomplete inhibition of P-gp by 40 mg/kg of WK-X-34 and the shorter action of WK-X-34. This would be the first report evaluating the effect of WK-X-34 on intestinal drug absorption by its oral administration. For effective and safe use of WK-X-34, further studies are needed to evaluate its appropriate dose in view of the influence on the metabolic enzymes and toxicity on mice.

In contrast to WK-X-34, the *AUC*_sys_ value of paclitaxel after oral administration in WT mice pretreated with LY335979 was approximately 2.9-fold higher than that in non-treated WT mice and comparable to that in *mdr1a/1b* KO mice. Moreover, LY335979 hardly affected the elimination process of paclitaxel. Dantzig et al. has demonstrated that the affinity of LY335979 for CYP3A4 is approximately 60-fold lower than that for P-gp [53]. This observation supports our results. However, the *F*_a_*F*_g_ value of paclitaxel after oral administration in WT mice pretreated with LY335979 was higher than that in non-treated *mdr1a/1b* KO mice, suggesting that LY335979 could inhibit CYP3A4 in small intestinal epithelial cells.

To further assess the effect of P-gp inhibitors on P-gp-mediated efflux of paclitaxel in mice without their influence on the metabolic process of paclitaxel, we carried out the measurement of plasma concentration of paclitaxel at the early phase after oral administration with a lower concentration of LY335979. The *C*_sys_ of paclitaxel at 10 min after oral administration in WT mice pretreated with P-gp inhibitors did not significantly differ from that in *mdr1a/1b* KO mice. These results indicate that the P-S difference method would achieve the evaluation of the effect of P-gp inhibitors on the intestinal absorption of paclitaxel with minimal influence on the metabolic process. In addition, we also observed that the *C*_sys_ and *C*_pv_ of topotecan at 30 min after oral administration in WT mice pretreated with LY335979 was almost as same as those in WT mice and *mdr1a/1b* KO mice. These results indicate that LY335979 hardly affects the BCRP-mediated intestinal drug absorption.

In conclusion, the present study has revealed that LY335979 would be the most valuable P-gp inhibitor for evaluating the sole contribution of P-gp to drug absorption. Using LY335979, we have also succeeded in the evaluation of the impact of P-gp on intestinal drug absorption without the influence on the metabolic process by using absorption rate. Since LY335979 inhibits P-gp function by allosterically interfering ATP hydrolysis [54], the present approach would be available for evaluating the contribution of P-gp to the intestinal absorption of various drugs, including BCS class IV drugs, without an increase in the dose of LY335979. On the other hand, multidrug resistance-associated protein 2 (MRP2) is expressed on the apical membrane of the intestinal epithelial cells as well as P-gp and BCRP, and therefore, further studies considering MRP2 are needed to demonstrate the selectivity of LY335979 for P-gp. Nevertheless, these findings make a valuable contribution toward evaluating the contribution of P-gp to drug absorption without using *mdr1a/1b* KO mice.

## Figures and Tables

**Figure 1 pharmaceutics-13-00388-f001:**
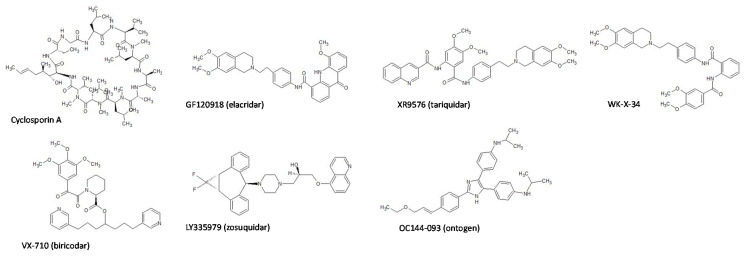
Chemical structures of cyclosporin A and P-glycoprotein P-gp inhibitors used in this study [20,28,29].

**Figure 2 pharmaceutics-13-00388-f002:**
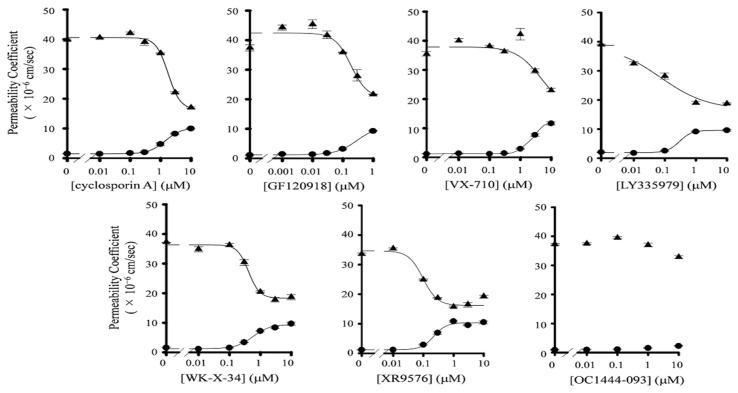
Inhibitory effect of P-gp inhibitors on P-gp-mediated efflux of paclitaxel in Caco-2 cells. The apical-to-basal (AP-to-BL) *P*_app_ value (*P*_app,AB_) (●), and basal-to-apical (BL-to-AP) *P*_app_ value (*P*_app,BA_) (▲) values were determined by the AP-to-BL and BL-to-AP transport of paclitaxel in Caco-2 cells in the presence or absence of various concentrations of P-gp inhibitors. The *P*_app,BA_ values of paclitaxel in the presence of GF120918, LY335979, and WK-X-34 were cited from our previous report [31]. Data are represented as mean ± S.D. for three experiments using different wells from a single passage of Caco-2 cells.

**Figure 3 pharmaceutics-13-00388-f003:**
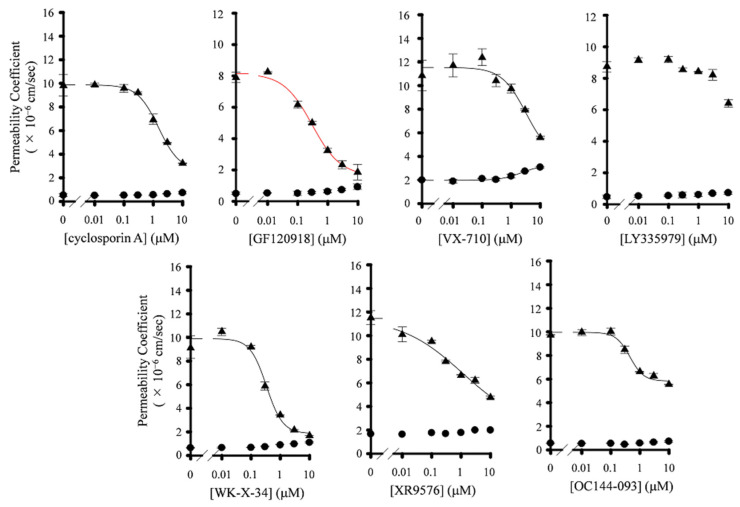
Inhibitory effects of P-gp inhibitors on BCRP-mediated efflux of mitoxantrone in Caco-2 cells. The *P*_app,AB_ (●), and *P*_app,BA_ (▲) values were determined by the AP-to-BL and BL-to-AP transport of mitoxantrone in Caco-2 cells in the presence or absence of various concentrations of P-gp inhibitors. The *P*_app,BA_ values of mitoxantrone in the presence of GF120918, LY335979, and WK-X-34 were cited from our previous report [31]. Data are represented as mean ± S.D. for three experiments using different wells from a single passage of Caco-2 cells.

**Figure 4 pharmaceutics-13-00388-f004:**
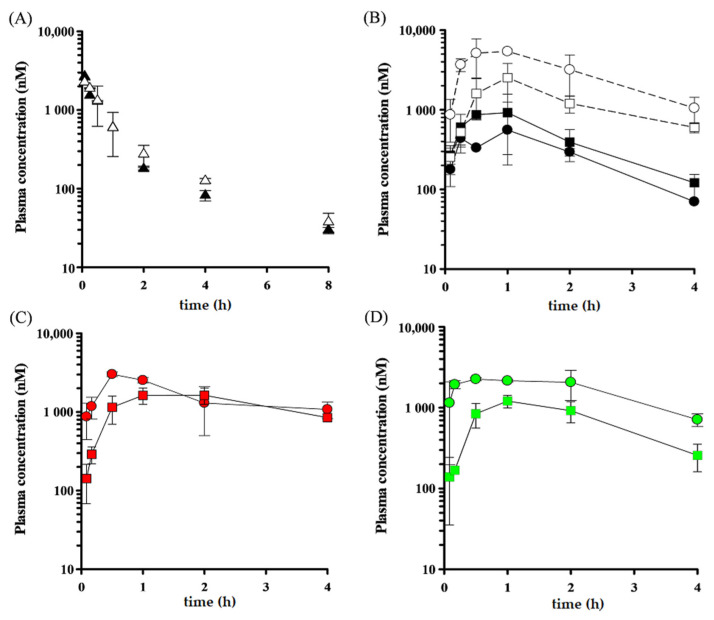
Plasma concentration-time profiles of paclitaxel after intravenous and oral administration with or without WK-X-34 and LY335979 pretreatment in mice. (**A**) Paclitaxel was intravenously injected into WT mice (▲) and mdr1a/1b KO mice (▵) at a dose of 5 mg/kg. (**B**) Paclitaxel was orally administered into WT mice and mdr1a/1b KO mice at a dose of 20 mg/kg. ●: portal plasma concentration in WT mice, ■: systemic plasma concentration in WT mice, ○: portal plasma concentration in mdr1a/1b KO mice, and □: systemic plasma concentration in mdr1a/1b KO mice. (**C**,**D**) Paclitaxel was orally administered into WT mice at a dose of 20 mg/kg, pretreated with 40 mg/kg WK-X-34 (**C**) or 60 mg/kg LY335979 (**D**). ●: portal plasma concentration, ■: systemic plasma concentration. Data are represented as mean ± S.D. (*n* = 3).

**Table 1 pharmaceutics-13-00388-t001:** Fifty percent inhibitory concentration (*IC*_50_) values and Hill coefficient of selected inhibitors on apparent AP-to-BL and BL-to-AP permeability, and efflux ratio (*ER*) of paclitaxel in Caco-2 cell monolayers.

Inhibitor	*IC*_50_ (nM)			Hill Coefficient		
	*P* _app,AB_	*P* _app,BA_	*ER*	*P* _app,AB_	*P* _app,BA_	*ER*
Cyclosporin A	1973 ± 21	1820 ± 126	502 ± 126	−1.63 ± 0.09	2.18 ± 0.39	1.63 ± 0.20
GF120918	319 ± 30	239 ± 97 ^a^	60 ± 21	−1.28 ± 0.09	1.63 ± 1.28	1.66 ± 0.51
XR9576	234 ± 61	64 ± 40	46 ± 19	−2.10 ± 0.64	18.4 ± 4.76	3.41 ± 0.28
LY335979	427 ± 52	107 ± 59 ^a^	115 ± 22	−2.23 ± 0.31	0.57 ± 0.35	1.71 ± 0.41
WK-X-34	935 ± 33	501 ± 132 ^a^	214 ± 113	−10.7 ± 6.00	1.29 ± 0.42	2.07 ± 1.22
VX-710	2680 ± 53	4496 ± 84	871 ± 277	−1.84 ± 0.07	1.08 ± 0.60	1.68 ± 0.38
OC144-093	n.c.	n.c.	n.c.	n.c.	n.c.	n.c.

^a^ Data from our previous report [31]. n.c., not calculated. *IC*_50_ on *P*_app,AB_: the inhibitor concentration to achieve 50% increase of *P*_app,AB_ of paclitaxel. *IC*_50_ on *P*_app,BA_: the inhibitor concentration to achieve 50% decrease of *P*_app,BA_ of paclitaxel. *IC*_50_ on *ER*: the inhibitor concentration to achieve 50% decrease of *ER* of paclitaxel (Appendix A). Data are represented as mean ± S.D. for three experiments using different wells from a single passage of Caco-2 cells.

**Table 2 pharmaceutics-13-00388-t002:** *IC*_50_ values and Hill coefficient of selected inhibitors on apparent AP-to-BL and BL-to-AP permeability, and *ER* of mitoxantrone in Caco-2 cell monolayers.

Inhibitor	*IC*_50_ (nM)			Hill Coefficient		
	*P* _app,AB_	*P* _app,BA_	*ER*	*P* _app,AB_	*P* _app,BA_	*ER*
Cyclosporin A	n.c.	2038 ± 13	1708 ± 248	n.c.	1.25 ± 0.17	1.33 ± 0.25
GF120918	n.c.	298 ± 13 ^a^	307 ± 23	n.c.	0.93 ± 0.41	0.92 ± 0.15
XR9576	n.c.	1000 ± 45	531 ± 162	n.c.	0.41 ± 0.08	0.44 ± 0.05
LY335979	n.c.	>10 μM ^a^	>10 μM	n.c.	n.c.	n.c.
WK-X-34	n.c.	370 ± 38 ^a^	328 ± 96	n.c.	1.56 ± 0.53	1.51 ± 0.41
VX-710	n.c.	2675 ± 31	1638 ± 520	n.c.	1.04 ± 0.40	0.46 ± 0.12
OC144-093	n.c.	n.c.	n.c.	n.c.	1.83 ± 0.59	3.61 ± 0.41

^a^ Data from our previous report [31]. n.c., not calculated. *IC*_50_ on *P*_app,AB_: the inhibitor concentration to achieve 50% increase of *P*_app,AB_ of mitoxantrone. *IC*_50_ on *P*_app,BA_: the inhibitor concentration to achieve 50% decrease of *P*_app,BA_ of mitoxantrone. *IC*_50_ on *ER*: the inhibitor concentration to achieve 50% decrease of *ER* of mitoxantrone (Appendix A). Data are represented as mean ± S.D. for three experiments using different wells from a single passage of Caco-2 cells.

**Table 3 pharmaceutics-13-00388-t003:** Pharmacokinetic parameters of paclitaxel after intravenous and oral administration to wild type (WT) and *mdr1a/1b* knockout (KO) mice.

		WT			*mdr1a/1b* KO			WT + WK-X-34		WT + LY335979	
		IV	po		IV	po		po		po	
			pv	sys		pv	sys	pv	sys	pv	sys
*Dose*	(mg/kg)	5	20		5	20		20		20	
*C* _max_	(nM)	---	730	442	---	2523	1179	3018	1631	2258	1208
*T* _max_	(h)	---	1	1	---	1	1	0.5	2	0.5	1
*AUC*	(nM·hr)	2320	2002	1089	2661	6575	3086	10,615	8636	7970	3201
*k* _e_	(h^−1^)		0.43	0.50		0.55	0.35	0.26	0.23	0.53	0.64
*t* _1/2_	(h)	2.38	1.62	1.37	2.12	1.25	2.00	2.69	2.98	1.30	1.09
*MRT*	(h)	1.71	2.51	2.29	1.47	2.16	3.09	4.15	4.67	2.42	2.25
*CL* _tot_	(L/h/kg)	2.52			2.20						
*Vd* _ss_	(L/kg)	4.32			4.15						
*BA*			11.7			29.0					
*F* _a_ *F* _g_			16.6			66.3		36.0		86.8	
*F* _h_			70.6			45.7					

*C*_max_: maximum plasma concentration; *T*_max_: time to maximum plasma concentration; *k*_e_: elimination rate constant; *t*_1/2_: elimination half-life; *MRT*: mean residence time; *F*_a_*F*_g_: apparent absorption ratio (*F*_a_) intestinal availability (*F*_g_); *F*_h_: hepatic availability.

**Table 4 pharmaceutics-13-00388-t004:** Absorption rate of paclitaxel (10 mg/kg) at 10 min after its oral administration with or without P-gp inhibitors in WT and *mdr1a/1b* KO mice.

	*n*	+Inhibitor (mg/kg)	*C*_pv_ (nM)	*C*_sys_ (nM)	*C*_pv_ − *C*_sys_ (nM)	Absorption Rate (*V*) (nmol/min/kg)
WT mice	3		439 ± 98	223 ± 76	216 ± 80	15.7
+Cyclosporin A	5	30	708 ± 77	263 ± 168	534 ± 102	38.7
+LY335979	6	30	1009 ± 149	202 ± 52	830 ± 140	60.2
+WK-X-34	3	30	746 ± 57	196 ± 64	549 ± 7.5	39.8
*mdr1a/1b* KO mice	6		967 ± 191	218 ± 100	749 ± 165	54.3

Data are represented as mean ± S.D. *C*_pv_: portal vein concentration; *C*_sys_: systemic circulation concentration.

**Table 5 pharmaceutics-13-00388-t005:** Absorption rate of paclitaxel (10 mg/kg) at 30 min after its oral administration with or without P-gp inhibitor in WT and *mdr1a/1b* KO mice.

	*n*	+Inhibitor (mg/kg)	*C*_pv_ (nM)	*C*_sys_ (nM)	*C*_pv_ − *C*_sys_ (nM)	Absorption Rate V (nmol/min/kg)
WT mice	3		328 ± 41	135 ± 41	165 ± 61	12.4
+Cyclosporin A	2	15	540	315	225	16.9
+LY335979	3	15	626 ± 82	368 ± 30	259 ± 45	19.5
+WK-X-34	3	15	674 ± 139	238 ± 63	437 ± 127	32.9
+Cyclosoprin A	3	30	1488 ± 146	829 ± 166	657 ± 38	49.4
+LY335979	3	30	1407 ± 70	524 ± 58	883 ± 59	66.4
+WK-X-34	3	30	2069 ± 204	757 ± 100	1311 ± 87	98.6
*mdr1a/1b* KO mice	3		1092 ± 118	245 ± 69	847 ± 144	63.7

Data are represented as mean ± S.D.

**Table 6 pharmaceutics-13-00388-t006:** Absorption rate of topotecan (2 mg/kg) at 30 min after its oral administration with or without P-gp inhibitor in WT, *mdr1a/1b* KO and *bcrp* KO mice.

	*n*	+Inhibitor (mg/kg)	*C*_pv_ (nM)	*C*_sys_ (nM)	*C*_pv_ − *C*_sys_ (nM)	Absorption Rate V (nmol/min/kg)
WT mice	3		196 ± 29	179 ± 25	17.1 ± 5.7	1.29
+Cyclosporin A	2	30	298	240	58.5	4.40
+LY335979	3	30	249 ± 26	228 ± 22.3	20.5 ± 9.7	1.54
+WK-X-34	3	30	464 ± 53	388 ± 19	76.6 ± 27	5.76
*mdr1a/1b* KO mice	3		236 ± 30	207 ± 28	28.9 ± 8.4	2.17
*bcrp* KO mice	3		469 ± 64	398 ± 88	70.5 ± 20	5.30

Data are represented as mean ± S.D. *bcrp*: breast cancer resistance protein.

## Data Availability

Data is contained within the article and its Appendix A.

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
