# Peer review of "Characterization of P-Glycoprotein Inhibitors for Evaluating the Effect of P-Glycoprotein on the Intestinal Absorption of Drugs"

_pharmaceutics, 2021, doi:10.3390/pharmaceutics13030388_

Round 1

Reviewer 1 Report

The aim of this paper was to investigate the effect of seven inhibitors of P-glycoprotein and their effects on improving the absorption of drug. The use of P-glycoprotein inhibitor can help better under the absorption of drugs when delivered orally and can also aid their absorption. The study uses Caco-2 cells in vitro and mouse models in vivo. Doing a head to head comparison of inhibitors in the same systems is a good idea and allows for better comparison. The in vivo work is very well done and proves compliments the in vitro data very well. 

Comments:

The expression of P-gp in Caco-2 cells has been shown to be variable. In addition, the substrates chosen in this study had only small increases in A to B absorption when an inhibitor was used. For this reason and to better validate the study if something like rhodamine 123 or digoxin and verapamil were used it would have been good (see below reference). Maybe you already have data that shows a low A to B absorption of a substrate that was increased more in the presence of an inhibitor. Another example would be Cyclosporin a with prazosin. I know you used paclitaxel to show specific Pgp effects. If you do have data showing this or expression levels of Pgp in your cells this data could be placed in supplementary.

Bentz, J., O'Connor, M. P., Bednarczyk, D., Coleman, J., Lee, C., Palm, J., Pak, Y. A., Perloff, E. S., Reyner, E., Balimane, P., Brännström, M., Chu, X., Funk, C., Guo, A., Hanna, I., Herédi-Szabó, K., Hillgren, K., Li, L., Hollnack-Pusch, E., Jamei, M., … Ellens, H. (2013). Variability in P-glycoprotein inhibitory potency (ICâ‚…â‚€) using various in vitro experimental systems: implications for universal digoxin drug-drug interaction risk assessment decision criteria. Drug metabolism and disposition: the biological fate of chemicals, 41(7), 1347–1366. https://doi.org/10.1124/dmd.112.050500

More explanation could be given for the inhibitors chosen in the introduction.

In the introduction verapamil is a described as a substrate and then as an inhibitor. A sentence is needed explaining how molecules can be substrate and inhibitor or it might be confusing for readers

Figue1 needs a reference

Section 2.2: Cell culture

Replace 3 weeks with number of days e.g. 21-14 days etc

References needed for concentrations/doses of inhibitors and substrates used.

Section 2.5: what type of anaesthesia was used ?

Section 2.8: Statistical analysis 

I was confused by this section. Understood that cells P55-70 were used and n=3 means 3 different passages but then it says “Here, data about in vitro transport experiments are represented as the mean ± standard deviation (S.D.) for 3 experiments using different well in a single passage (P55) of Caco-2 cells”. Is this not an  n of 1 ? All 3 were carried out on the same passage.  Does this mean all experiments were carried out at P55 ?  More variability is needed such as using more than one passage (e.g. biological replicates). This section needs to be clarified. 

The information on passage of cells used could be include in section 2.2 cell culture.

Table 1: this table needs to be improved, I found it confusing. Are these the papp values for each IC50 values for each inhibitor ? And teh ER shown for what concentration ? 

Reviewer 2 Report

Abstract

 In this study, we determined the comparative analysis of the inhibitory activities of seven P-gp inhibitors (cyclosporin A, GF120918, LY335979, XR9576, WK-X-34, VX-710, and OC144-093) to evaluate the effect of P-gp on drug  absorption. GF120918, LY335979, and XR9576 significantly decreased the basal-to-apical transport  of paclitaxel, a P-gp substrate, across Caco-2 cell monolayers. GF120918 also inhibited the basal-to- apical transport of mitoxantrone, a breast cancer resistance protein (BCRP) substrate, in Caco-2 cells,  whereas LY335979 hardly affected the mitoxantrone transport. In addition, the absorption rate of  paclitaxel after oral administration in wild-type mice was significantly increased by pretreatment  with LY335979, and it was similar to that in mdr1a/1b knockout mice. Moreover, the absorption rate of topotecan, a BCRP substrate, in wild-type mice pretreated with LY335979 was similar to that in mdr1a/1b knockout mice but significantly lower than that in bcrp knockout mice. These results indicate that LY335979 has selective inhibitory activity for P-gp, and would be useful for evaluating  the contribution of P-gp to drug absorption.

General comments

Interesting paper and well conducted investigation. Due to the rather complicated experimental design, the authors are invited to implement the few lines that are dedicated to the aim of the study, by illustrating in reasonable details the experimental plan. Three model drugs are cited but it is not clear   which ones are  used in which experiment and why, either in vitro or in vivo.

Concerning the experimentals, the choice of the concentrations for both drugs and inhibitors, whenever relevant, should be commented. Simirarly the choice of the doses in animal studies should be commented.

The high dose ratios between inhibitor and drug substance should also deserve a comment, in vew of exploiting the inhibition approach to allow for oral administartion of class IV problematic drugs.

As for in vivo experiments two pieces of information are missing in the experimentals: the number of mice used in eachand every experiment and the sampling protocol (pooled samples?), as well as the PK software used for pharmacokinetic analysis.

For easing readers' understanding, the discussion section should be more profitably subdevided into sections.

Reviewer 3 Report

Yusuke Kono et al investigated in vitro and in vivo studies of Pgp inhibitors to identify the selective pgp inhibitor for drug absorption. Authors well designed the studies using CaCO2 cells and WT and KO mice for evaluating Pgp inhibitors.

  1. Authors should check the manuscript for spelling mistakes and need to correct them. Cyclosporine A needs to be corrected in tables 4, 5, 6. Abbreviation CsA needs to be used throughout the manuscript to minimize mistakes.

AUC, AUMC are in different formats and need to be corrected throughout the manuscript.

Mdr1a/1b spelling is wrongly mentioned in table 4 and should be corrected.

Mean±S.D. (not means±S.D.). It should be corrected in all figure legends, figure and table footnotes, supplementary figure legends.

  1. Include the source information of solutol, fetal bovine serum, nonessential amino acids, glutamine.
  2. Include the information on the age of animals, temperature, and humidity conditions of the animal facility room. Information about the wild type mice is missing in the 2.3 animals section and should be included.
  3. The authors need to mention the anesthesia procedure in in vivo methods section.
  4. Mass balance (% recovery) details are not mentioned in the data analysis section.
  5. Did the authors use any software for pharmacokinetic data analysis?
  6. There are no supplementary Tables S1 and S2 in the manuscript. Hill coefficient values were not mentioned in Tables 1 and 2. Include the hill coefficient data in the tables.
  7. What is the effect of substrates and inhibitors on CaCO2 cell viability and on mice?
  8. The concentration-dependent inhibitory effects of LY335979, WK-X-34, GF120918 on the basal to apical transport of paclitaxel and mitoxantrone in CaCO2 cells were already published in the article: “The Impact of Breast Cancer Resistance Protein (BCRP/ABCG2) on Drug Transport Across Caco-2 Cell Monolayers”. In figures 1 and 2, how the concentration-dependent inhibitory effect of these three pgp inhibitors is different from the published article. The Papp BA IC50 values of these three inhibitors mentioned in tables 1 and 2 are matching with the IC50 values mentioned in the published article. The authors should clarify how the published data are different from the manuscript data. Authors should discuss the published data and cite this published article.
  9. Why the authors not used GF120918 instead of WK-X-34 in in vivo mice studies? For dual inhibitor, GF120918 is more potent than WK-X-34 based on the CaCO2 data.
  10. Line 279, 299, 309, 319, 320: In the 3.3 section, the pharmacokinetic parameter values are not matching with the numbers mentioned in table 3. The authors should modify the information mentioned in the 3.3 section.
  11. The Ke values are missing for WT mice. Include the values in table 3 and at line 306.
  12. In figure 4 and table 3, the authors should include the number of animals used for these studies. In table 3, mean±SD data are missing and should include the mean±SD data for all pharmacokinetic parameters.
  13. In figure 4 legend, modify the sentence “data are represented as mean+SD for 3 experiments” related to animal experiments.
  14. What is the pretreatment duration for pgp inhibitors in in vivo studies?
  15. Why the number of mice used for the cyclosporine A treated group is low in Tables 5 and 6?  

Round 2

Reviewer 1 Report

Thank you for addressing all issues that were raised. 

For cell culture you could say "Here, data about in vitro transport experiments are represented as the mean ± standard deviation (S.D.) for 3 experiments using Caco-2 cells with different passage with 3 replicates per n". 

This is just a suggestion.

Reviewer 3 Report

Yusuke Kono et al significantly improved the manuscript in the revised version.

  1. In Table 1, SD values are not mentioned for the Hill coefficient of Papp AB values and need to be included.

2. It is not clear how the pharmacokinetic studies and blood collection were performed in figure 4 and table 3. This method section should be properly described for clear understanding. How much blood volume was collected for each time point? How many time points were collected from each animal? Does it include the sparse sampling collection method used for the PK studies? How much total blood volume was collected from each animal? As per animal research and ethics, the maximum blood volume that can be drawn is 10-15% (200-300 µL) from each animal. It is not possible to collect all time points from the portal and abdominal veins of each animal in these studies.  
